# COVID-19 and Policy-Induced Inequalities: Exploring How Social and Economic Exclusions Impact ‘Temporary’ Migrant Men’s Health and Wellbeing in Australia

**DOI:** 10.3390/ijerph20136193

**Published:** 2023-06-21

**Authors:** Daile Lynn Rung

**Affiliations:** Freemasons Centre for Male Health and Wellbeing, Faculty of Health, Charles Darwin University, Casuarina 0810, Australia; daile.rung@cdu.edu.au

**Keywords:** temporary migrants, migrant men, men’s health, wellbeing, social exclusion, economic insecurity, COVID-19, Australia, policy, welfare, work rights

## Abstract

The Australian government swiftly put in place a number of economic relief measures and policies to support people during the COVID-19 crisis. However, the government’s COVID-19 response policies excluded people with ‘temporary’ migrant status living in the country and encouraged people holding temporary visas who lost jobs and could no longer afford to support themselves to ‘go home’. This paper draws upon sub-citizenship theory to explore how Australia’s immigration and COVID-19 response policies are likely to impact the health and wellbeing of ‘temporary’ migrant men and their families. Through focusing on Australia’s policy approach towards ‘temporary’ migrants and the social, health, and human rights implications among men with temporary migrant status during the pandemic, this paper contributes to emerging literature that considers the intersectional implications of immigration and COVID-19 response policies as they pertain to people with precarious migration status. Applying sub-citizenship theory to analyse how Australia’s COVID-19 response policies intersect with ‘temporary’ migration schemes offers a useful way to think about and unearth how structural, and often legislated, exclusions can affect the health and wellbeing of marginalised groups.

## 1. Introduction

Since the COVID-19 pandemic emerged in December 2019, communities around the world have faced immense stressors upon their health and wellbeing. The pandemic has brought to the forefront structural vulnerabilities faced by marginalised groups and individuals [1]. Globally and in Australia, people with temporary migrant status have emerged as one of the most vulnerable groups to experience health and wellbeing inequalities associated with COVID-19 [2,3,4,5].

As the coronavirus infection spread, public health directives in most countries began to call for social distancing along with various policies and measures, including border lockdowns aimed at slowing down the spread of the virus. In its bid to pursue a zero-COVID suppression policy aimed to minimize community transmission of COVID-19, Australia closed its borders to all non-residents on 20 March 2020 and only fully opened its borders to international visitors in February 2022. Throughout 2020 and 2021, different states and territories within Australia were implementing lockdowns and social restrictions to various degrees to try to curb community transmission [6]. In Australia, this strategy, known as ‘flattening the curve’, was supported by the government through physical distancing policies and economic support packages, known as Jobkeeper and Jobseeker [7]. However, the government’s economic support policies in response to COVID-19 did not make support available to all people living in Australia. From the outset of the COVID-19 pandemic, ‘citizens’ and ‘permanent residents’ were prioritised, while ‘temporary’ migrants were explicitly excluded [2,3,4,8,9,10,11]. The Australian government rationalised this exclusion as needing to ‘draw a line somewhere’ [12] and encouraged people holding temporary visas to ‘go home’ if they cannot economically support themselves [13].

There are an estimated 1.9 million temporary visa holders in Australia as of June 2022 [14]. Temporary migrants are a diverse group of people who hold temporary visas for work, education, and family purposes in Australia. Temporary migrants include international students, New Zealanders on Special Category visas, working holiday makers, skilled and low-skill workers, post-study graduate visa holders, and partner and family visa holders. In 2022, the top three nationalities of all temporary visa holders in Australia included nationals from New Zealand (34.6 per cent), India (14.7 per cent), and China (7.3 per cent) [14]. Although not officially counted as ‘temporary’ migrants, another substantial group of people who may hold temporary visas include refugees and asylum seekers on temporary protection and bridging visas.

Unfortunately, as people with temporary migrant status are without voting rights, there is little political will to explore and understand how their social and economic conditions impact their health and wellbeing. Scholarship is needed to explore how exclusionary social policies might affect the health and social and emotional wellbeing among different sub-sets of temporary migrants as they comprise a large and diverse group. To contribute to this work, this paper explores how COVID-19 policy exclusions have compounded pre-existing social and economic exclusions faced by men living in Australia with temporary migrant status.

Drawing upon sub-citizenship theory [15,16], I aim to explore how ‘temporary’ migrant men’s health and wellbeing may be affected by an array of pre-existing social and economic exclusions, which were compounded by exclusionary social and economic policies in response to the pandemic. Concentrating on the role of exclusionary policies and social practices in the context of COVID-19, I intend to analyse the health and wellbeing implications of men living in Australia with temporary migrant status and, by extension, those of their families and children. To accomplish this, I consider how pre-existing vulnerabilities and exclusions from work, welfare supports, and free public services, including healthcare and education, are likely to affect the health and wellbeing of ‘temporary’ migrant men and their families.

People with temporary migrant status in Australia have been excluded from accessing public services and welfare safety nets for many years, long before the global pandemic. However, as people holding temporary visas in Australia were uniformly excluded from accessing the government’s roll out of COVID-19 support measures, it is important to ascertain how the compounding of exclusionary immigration, public health, and pandemic response policies pre- and post-COVID may affect the health and wellbeing of temporary migrant men and their families in the country.

There is increasing consensus among researchers investigating the impact of the virus from medical, public health, and sociological perspectives that intersectional analyses are urgently needed to create more equitable pandemic policies and interventions [17,18,19]. Intersectional approaches that analyse how biological sex, sociological aspects of gender, and diverse social inequalities affect people’s health and wellbeing are helpful to build understandings of how COVID-19 affects diverse groups of people in different ways.

Addressing the nexus between COVID-19, equity, and men’s health from a public health perspective, Smith et al. [17] note: the vulnerability of certain groups are rarely acknowledged in popular and social media in comparison to broader population-wide discussions about COVID-19; and seldom have these recognised the inequalities faced by men (p. 51).

Through focusing upon men with temporary migrant status in Australia, this paper contributes to emerging scholarship that explores “how future public health pandemic approaches could better respond to the needs of the most vulnerable and marginalised groups of boys and men in a timely way” [17] (p. 49). The economic and social impact of COVID-19 policies upon migrant men, who are often the breadwinners in migrant households, will undoubtedly have health and wellbeing implications for migrant fathers and their families.

Many of the issues raised in this paper focused on men with temporary migrant status also apply to women, children, LGBT communities, and undocumented migrants, albeit in complex and varied ways. This paper’s focus on men holding temporary visas should not be taken as a statement that they face more structural inequalities than women and LGBT communities who hold temporary visas or other groups of migrants, including undocumented migrants.

While this paper’s intersectional approach focuses on men with temporary migrant status, it acknowledges that certain sub-groups of temporary migrant men, including LGBT migrants, experience compounded forms of social, economic, and health inequalities [20]. It must also be acknowledged that certain groups of migrants, including undocumented migrants, are more at risk of experiencing social, economic, and health inequalities. For example, Australia’s punitive immigration detention regime confines undocumented migrants in close quarters in detention centres where they faced extremely high risks of COVID-19 infection and possibly death [21]. Viewing this issue through a sub-citizenship lens, undocumented migrants are institutionally positioned at lower levels of the sub-citizenship hierarchy, making them more vulnerable to state-led violence and subordination.

This critical social policy analysis paper has two interrelated aims. First, it explores how Australia’s neoliberal approach to immigration creates social and economic exclusions and barriers for all people who hold temporary visas in the country. Second, building upon this key insight, it considers how Australia’s COVID-19 response policies have deepened pre-existing social and economic exclusions that are likely to present certain health and wellbeing challenges for men with temporary migrant status and, by extension, their families.

Two broad threads of literature are drawn upon to provide the necessary contextual background to explore ‘temporary’ migrant men’s health and wellbeing in the context of COVID-19 in Australia. First, literature pertaining to COVID-19 and men’s health suggests that men are a highly vulnerable group with respect to COVID-19. Second, I draw upon literature and public policies in Australia focused on temporary migrants’ health and wellbeing pre- and post-COVID. The paper sheds clarity around the notion of ‘temporary’ migrants through discussing how it has come to be that millions of people hold this precarious migration status in Australia and elsewhere.

To operationalise the critical social policy approach, I apply an intersectional sub-citizen lens to explore how Australia’s COVID-19 policy responses multiplied and compounded pre-existing economic and social exclusions faced by people with temporary migration status. Sub-citizenship theory [16] enables a deep consideration of how the predominance of neoliberal dynamics underpinning immigration and public policies works to sort and categorise people to experience different levels of subordination and inequalities largely based upon the allocation of migration and citizenship status. Through a sub-citizenship approach, the focus is on how it has come to be that pre-existing exclusions and vulnerabilities with respect to work and access to welfare supports and public services have become compounded for those with temporary migrant status and how such exclusions are likely to result in adverse health and wellbeing among temporary migrant men.

The premise underlying this paper is that COVID-19 public health and pandemic response policies should have the end goal of improving the health and wellbeing of all people in society regardless of their gender, race, age, migration/citizenship status, socioeconomic status, sexual orientation, or any other attribute. The United Nations’ Sustainable Development Goals (SDG) can be advanced through inclusive public health, immigration, and social policies that promote good health and wellbeing (SDG 3) and reduce inequality (SDG 10) [22]. With these goals in mind, this paper aligns with scholarship advocating for a gendered and human rights lens in public health and pandemic research and policy responses.

## 2. COVID-19 and Men’s Health

The available evidence suggests that being male is a key factor determining the health and wellbeing outcomes of people who contract COVID-19. Sex-disaggregated epidemiological analyses across different parts of the world reveal a gender gap associated with the virus [23].

Globally, epidemiological findings indicate that men are more vulnerable to COVID-19 than women, with men experiencing higher morbidity and mortality rates [23,24]. Indeed, most countries with available data indicate a higher male-to-female case fatality ratio [25]. Data from China indicate that the majority (66.7%) of COVID-19 patients have been male. In Italy, men accounted for 58% of infected patients and 70% of COVID-19-related deaths. In the United States, 53.5% of reported COVID-19 deaths have been in men [26].

At the present time, it is too early to tell what accounts for this gender gap in COVID-19-related deaths, although some have speculated it may be related to sex-based immunological differences between males and females; gendered lifestyle behaviours, such as higher rates of smoking and drinking; and less adherence to preventative measures, such as handwashing and mask-wearing, that may put men more at risk [23]. Most of the available literature exploring the COVID-19 gender health gap has been bio-medically focussed on the possible role that biological sex differences, such as hormones and chromosomes, may play with health and wellbeing differences between men and women.

There has been less attention focused on the underlying social and contextual factors that may shape men and women’s health outcomes with respect to COVID-19 [26,27]. Focusing exclusively on decontextualised statistics about sex differences is dangerous because it obscures other factors that may be equally—as or perhaps more—relevant than biological sex in shaping people’s vulnerability to the virus itself and the knock-on social and economic consequences associated with the global pandemic. Sociological studies offer the benefit of adding much-needed contextual analyses through critically analysing the factors that connect individuals to their social landscape. Thus, there is a great need for sociological approaches, as they allow us to better understand the underlying factors that may be accounting for the gender gap observed in the rich epidemiological data already available. Investigating how different groups of people were and continue to be affected in a post-COVID world would benefit from intersectional approaches that take account of how gender, sex, age, race, socio-economic status, and other markers of difference make some groups more vulnerable to certain health and wellbeing challenges.

While we know that COVID-19 has negatively impacted men’s health and wellbeing in a general sense, little information is available about how the health and wellbeing of migrant and refugee men’s has been impacted. Borgkvist [28], Adamson and Smith [29], and Rung and Adamson [30] note there is limited research on groups of marginalised fathers, including migrants and refugees. Internationally and in Australia, there is a critical gap in knowledge about ‘temporary’ migrant men’s mental health and wellbeing in general and in the context of COVID-19. To address this gap, this paper contributes to sociologically oriented literature to understand how complex social and economic factors may impact the health and wellbeing of ‘temporary’ migrant men and, by extension, their families living in Australia in the aftermath of COVID-19.

Most of the available data that pertain to migrant men’s health and wellbeing in the Australian context draws upon the Longitudinal Study of Australian Children (LSAC), which excludes people with temporary migrant status. However, as the LSAC includes people from migrant and refugee backgrounds who have Australian citizenship or permanent residency status, it provides some indication of the health and wellbeing challenges faced by migrant men who have more secure migration and citizenship statuses. Men with less secure migration and citizenship statuses may have similar (and likely more complex) health and wellbeing challenges as migrant men who hold permanent residency visas or are citizens. However, to date, no studies have focused upon the health and wellbeing of men with temporary migrant status pre or post pandemic.

Overall, a small number of studies that draw upon LSAC data suggest that migrant and refugee men with more secure residency and citizenship status are at risk of poorer mental and general health, particularly when they face employment challenges. Whilst these studies draw upon LSAC data that pre-dates the emergence of COVID-19, these findings show that migrant and refugee men in Australia are more vulnerable to health and wellbeing challenges and job conditions have a strong impact upon the health and wellbeing of migrant men.

Considering that the social and emotional wellbeing of fathers from migrant and refugee backgrounds is strongly linked to fulfilling what they view as their cultural role as financial and figurative providers [30], there is a need to explore how certain employment and financial challenges associated with COVID-19 may affect the health and wellbeing of ‘temporary’ migrant men and their families in Australia.

However, to shed some clarity around the highly misleading term, ‘temporary’ migrant, it is useful to first understand how it has come to be that millions of people with temporary migrant status live in Australia and why they are institutionally positioned as a highly vulnerable group of people.

## 3. ‘Temporary’ Migrants in Australia

In Australia, people are lawfully classified into three general categories: citizens, permanent residents, and short-term visitors. Australia’s immigration system divides migrants into two main categories: permanent migrants, who hold visas to live in Australia indefinitely, and temporary migrants, whose visas have time limitations.

There are three important points to keep in mind about people living in Australia with temporary migrant status. First, temporary migrants represent a significant population in Australia, with nearly 2 million temporary migrant visa holders in the country [31]. Second, temporary migrants can—and increasingly do—live long-term in Australia [16,32,33,34,35]. Third, temporary migrants are highly diverse in terms of gender, race, socio-economic status, age, and sexuality.

While ‘temporary’ implies that this group lives in a country on a short-term basis, ‘temporary’ migrants can—and increasingly do—live long term in Australia [16,32,33,34,35]. Mares’s (2016) [33] notion of ‘long-term temporary migrants’ is useful as it denotes that people often hold this precarious migration status for many years. It is most useful and accurate to view the term ‘temporary’ migrant a *status* rather than an indicator of how long a person has actually lived in and contributed to society [15,16,33]. Australian immigration policy changes over the past thirty years have resulted in millions of long-term temporary migrants [16,33].

Hugo (2014) [35] notes that the growth of temporary migration is one of the most significant developments to the dynamics of Australian immigration policy. In many countries, including Australia and Canada, there has been a sharp increase in temporary migrant schemes at the expense of both permanent immigration and humanitarian resettlement programs [36]. Over the past few decades, policy changes to the composition of immigration pathways have fundamentally changed Australia from the settler society model of the 20th century into what is now arguably best characterised as a temporary migrant society [15,33,35]. In fact, since 2000, temporary migrants have far outnumbered permanent ones [37].

Millions of migrants holding temporary visas live in Australia long term, work, study, pay taxes, obey laws, form social attachments, develop feelings of home and belonging, and contribute to the social, economic, and cultural life of the nation. However, due to their status, this group of people lacks access to a range of public services and benefits—including healthcare, education, and social safety nets—and are denied voting rights [33]. However, it should be noted that ‘temporary’ migrants in Australia are considered residents for tax purposes only [38]. This means that all public services, welfare supports, Medicare, and other social and economic safety nets are funded through the tax contributions of a group of people who are deemed ineligible to access free public services, supports, and safety nets in times of need. People holding temporary visas in Australia usually must pay for all services, such as healthcare and education for their children, until they acquire permanent resident and/or citizenship status. In recent years, immigration policy changes have required permanent residents to serve a ‘waiting period’ before they can access Australia’s income support system [39]. As of July 2021, the government announced that permanent residents would have to wait four years to access most welfare support services. In the previous policy, the waiting period was two years. It should be noted that time spent in the country as a ‘temporary’ migrant does not count towards the waiting period.

While most people with temporary migration status in Australia have some degree of working rights and pay taxes, they are more at risk of experiencing exploitation and other forms of mistreatment in the workplace [40]. As ‘temporary migrants do not stand on the same firm legal ground as citizens and permanent residents’ [33] (p. 6) they run a far greater risk of experiencing exploitation and abuse in the workplace.

Welfare restrictions contribute to economic insecurities among temporary migrants, particularly if they lose their employment. Being excluded from welfare and other supports puts temporary migrants in a highly precarious economic position, which in turn makes them more willing to do the ‘dirty, difficult and dangerous jobs that nationals will not’ [40] (p. xiii). Without access to freely available public services and safety nets, temporary migrants are more vulnerable to underpayment, wage theft, superannuation theft, and other forms of mistreatment as they have limited supports and alternatives to leave unscrupulous employers and abusive workplace conditions.

This paper acknowledges that the barriers to social, political, and economic participation currently confronting ‘temporary’ migrants is not an accidental or unlucky occurrence, but rather should be viewed as “a direct consequence of dramatic and deliberate changes in immigration policy since the 1990s” [3] (p. 62). The expansion in both the number of ‘temporary’ migrants and the elongation of time that many people occupy this status has come about through deliberate changes to Australian immigration and citizenship policies. This begs the question, “What accounts for the policy changes that have increased the number ‘temporary’ migrant and the length of time they hold this precarious status in Australia?”.

## 4. Temporary Migration Schemes and Neoliberal Citizenship

Neoliberal economic doctrines that first emerged in the 1970s and came into prominence in the 1980s have had an immense influence upon how nation-states retool immigration and citizenship policies [41]. Under neoliberal economic theory, everything has a ‘market value’, including social belonging vis à vis access to immigration and citizenship pathways. Australia is not the only country whose approach to immigration and citizenship policy has been retooled in neoliberal terms, as this global trend towards commodifying immigration and citizenship policies has become particularly apparent in affluent, western countries. Australia’s immigration system and its militarised, securitised approach to border enforcement is arguably one of the most extreme manifestations of neoliberal citizenship in the world [15,16,42].

Neoliberal citizenship is characterised by “a contractual view that sees citizenship no longer primarily as a prima facie right but as a prized possession that is to be earned and can be lost if not properly cultivated” (p. 408) [43]. Through reframing citizenship in an increasingly contractual and contingent manner, the nation-state provides the institutional machinery necessary to redraw and reengineer the boundaries and tenor of citizenship though the creation of market-driven policies [41]. Under neoliberal terms, citizenship and the rights associated with it are no longer regarded as human rights connected to dwelling in a territorial space and being a member of a particular society, but as commodified, contractual rights [41,43].

Neoliberal citizenship is marked by progressively tightening people’s access to even the most basic services, safety nets, and human rights [16,44,45,46,47]. A marker of the advance of neoliberal citizenship is found in reframing access to ‘public’ services, welfare, and social safety nets as only being available people who the nation-state deems as ‘belonging’, based upon migrant and/or citizenship status. Neoliberal approaches to citizenship and immigration pathways tend to work against the mobility, security, and human rights of people who have less-than-full status in their county of residence (i.e., ‘non-citizens’) [16]. Those whom the nation-state construes as not belonging to its imagined community [48] face compounded, and often legislated, exclusions. Increasingly, access to welfare support in many countries is only available to those deemed ‘deserving’, often based their demonstrated ability to work in the formal economy.

One of the main ways that neoliberal citizenship is being realised in Australia is through the expansion of temporary migration schemes. As the nation-state shifts the rules, pathways, requirements, and monetary commitments required for immigrants to attain more secure migration and citizenship statuses, the result has been more long-term temporary migrants. Castles (1995) [49] warns that immigration policy models that bar people from attaining permanent settlement creates ‘differential exclusion’ among migrants and stands in direct opposition to the democratic principle of including all members of society. Temporary migrant schemes are neoliberal policy models par excellence due to their ability to incorporate immigrants into certain areas of society, above all the labour market, whilst excluding millions of people from accessing welfare and social safety nets, permanent residency, and citizenship rights, including political participation.

Australian immigration intakes are increasingly displaying a preference for ‘staggered’ or ‘two-step’ pathways towards permanent residence [33,50]. In 2012–2013, less than half (40.2%) of permanent visas were obtained onshore i.e., people who lived in the country as ‘temporary’ migrants [51]. By comparison, in 2019–2020, nearly two-thirds (64.5%) of permanent residence visas in Australia were granted to people through the onshore program [52].) As people who obtain permanent residency status are predominately former temporary migrants, this trend suggests that the regulatory processes governing Australia’s immigration system have been retooled to allow for a more drawn-out immigration process. Extending temporary migration pathways is highly lucrative for nation-states and immigration intermediaries, yet highly precarious for people with temporary migrant status [16,36,50,53].

Immigration pathways leading from temporary residency to permanent residency to citizenship are far from certain. The longer people are construed as ‘temporary’ migrants, the more precarious their lives become as they do not have access to the same social protections and legal footing as permanent residents and citizens. Not having access to welfare supports and services can be detrimental to people who experience normal life events such as sickness, separation, domestic violence, pregnancy, raising children, or losing a job. When the global pandemic emerged in Australia in early 2020, temporary migrants were among the most vulnerable due to their precarious social status in the country.

In recent years, it has become increasingly common for temporary visa holders to go through numerous temporary visas, where they run the risk of becoming long-term and potentially indefinite temporary residents [33]. As Mares [33] explains, temporary visa holders are in danger of “moving around in circles, jumping repeatedly from one temporary visa to another. If they cannot leap to a safe landing before their temporary visa options are exhausted, then they are out” (p. 32). Elongated and staggered immigration policy approaches ensure that ‘temporary’ migrants contribute to the economy, yet most will not find a pathway towards permanent residence and the social, economic, and political rights associated with that status.

## 5. Viewing ‘Temporary’ Migrants through a Sub-Citizen Lens

Sub-citizenship theory provides an open-ended and relational approach to explore how people’s experiences with social, political, and economic exclusions and access to human rights are connected to institutionalised processes governing migration and citizenship. In earlier work, I defined sub-citizenship as “translocal processes of subordination that create various hierarchal conditions of precarity and dehumanization for different groups of people primarily based upon, but not wholly determined by, migration and citizenship status” [16]. Sub-citizenship is structurally produced and enacted though immigration policies and processes that sort, classify, and assign people into a hierarchical array of migration and citizenship statuses. This process is underpinned by nation-states’ monopoly on classifying people as belonging to one of two constructed binaries: ‘citizens’ and ‘non-citizens’ [16].

This paper’s use of sub-citizenship theory focusing on the health and wellbeing implications among temporary migrant men in the context of COVID-19 policy exclusions expands upon previous approaches to sub-citizenship that have analysed how policies and practices led to the subordination and dehumanisation of other groups of migrants including children with illegalised migrant status in Australia and America [16], the UK’s ‘Windrush scandal’ [54], migrant workers and Australia’s COVID-19 governmental responses [4], the conflation of globalisation and migration [55], integration and transnational practices of New Zealand migrants in Australia [56], and alternatives to immigration detention in Indonesia [57].

Structurally linked to the neoliberal market paradigm, sub-citizenship is a process that creates and draws upon disempowered migrant labour to expand markets and prop up the empowered labour, wages, and social protections enjoyed by people with more secure migration and citizenship status, i.e., citizens and permanent residents [16]. Nation-states create and sustain the legal frameworks, policies, and practices that perpetuate and expand sub-citizenship though crafting boundaries that differentially exclude [48] certain groups of people socially, politically, and economically.

While primarily aimed at expanding the global capitalist economy, sub-citizenship subjects all people to various forms of social, political, economic, and territorial expulsion [16]. Those positioned at lower levels of the sub-citizenship hierarchy are more vulnerable to social and economic exclusion, exploitation, state-led violence, and human rights abuses. The imposition of torture and indefinite detention without the right to habeas corpus is increasingly deployed against those positioned at the bottom of the sub-citizen hierarchy, i.e., people with illegalised migration statuses [16,58].

Within sub-citizenship’s hierarchical structure, ‘temporary’ migrants are institutionally positioned as a highly vulnerable group of ‘non-citizens’ due to their precarious, or insecure, migration status. Living in countries, often long-term, with less than full status puts temporary visa holders at risk of numerous and overlapping social, political, economic, and human rights exclusions [36].

Under Australia’s current immigration policies, ‘temporary’ migrants:lack the right to permanently reside in the country;have partial work rights, which are temporary in nature;are often dependent on a third party (such as an employer who acts as their ‘sponsor’) to secure their employment and residency rights;do not have access to free public services, such as healthcare and educationlack access to welfare and social protections, including COVID-19-related pandemic responses (Jobseeker and Jobkeeper);lack voting rights; andare at risk of being deported or detained if they breach their visa conditions [4,9,16,33,36].

As nation-states retool their immigration and citizenship policies, temporary migrant status often becomes a mechanism that bars certain people from accessing welfare and public services, such as healthcare and education, while at the same time ensuring that this differentially excluded group serves as a net contributor to the public purse.

## 6. Australia’s COVID-19 Policy Impacts upon ‘Temporary’ Migrants

In Western countries, the most vulnerable populations tend to be people without permanent legal status, such as asylum seekers and temporary visa holders, and other poor, marginalised citizens with limited or no access to healthcare. People with temporary migrant status have long been among the most marginalised populations around the world, as they have limited (and often no) access to public health care, social protection, work rights, public education, permanent residency, and voting rights [3,9,16,36,59].

Through an intersectional, sub-citizen lens, we can develop better understandings of how Australia’s immigration and COVID-19 response policies sorted, classified, and excluded certain groups of people based on the citizen/non-citizen construct. The sections below explore how ordering people into migration classifications and legal statuses exacerbated pre-existing social and economic inequalities among temporary migrants who were excluded from Australia’s pandemic assistance schemes based on their low positioning within the sub-citizenship hierarchy.

## 7. Heightened Job Insecurities

Predating the emergence of COVID-19, people with temporary migrant status in Australia faced multiple and overlapping vulnerabilities and exclusions from accessing fair work, including exploitative and unsafe working conditions [40,60] with wage theft and underpayment endemic in occupations and industries that employ temporary migrant workers [61,62,63]. In 2018, a survey of 1433 international student visa holders found that 100% of those working in the restaurant and retail sectors were underpaid [62]. Clibborn (2018) [62] speculated that the international students in the study tolerated the underpayment because it was viewed as a widespread and normalised behaviour among this disenfranchised group of workers.

The pandemic undoubtedly intensified pre-existing economic insecurities and workplace vulnerabilities among people with temporary migration status in Australia. In early 2020, Australia began to enforce social distancing and lockdowns to contain the spread of COVID-19. As a result of these policies, certain sectors, such as tourism, retail, and restaurants, and the informal economy were among the first to shed jobs. Temporary migrants, who are concentrated in heavily casualised and informal sectors of the economy, were among the first to lose their jobs during the government’s COVID-19 lockdowns. As many temporary visa holders, most notably international students, often do not have full work rights, they are more economically vulnerable if they experience job loss or reduction of work hours. While the 40 h per fortnight limitation placed on work hours was relaxed for international student visa holders working in essential services such as supermarkets and aged care [64], this move was not enough to combat wide-spread job losses and economic insecurities experienced by this marginalised group of workers.

To date, one large-scale study has focused on how Australia’s COVID-19 response policies impacted temporary visa holders living in the country. In July 2020, over 6100 temporary migrants in Australia were surveyed, including over 5000 international students and further thousand temporary migrant visa holders including Working Holiday Makers, Temporary Graduate visa holders, Temporary Skill Shortage visa holders, refugees, and people seeking asylum [2]. Key findings from the study include temporary migrants reporting:a critical loss of income from job loss and/or diminished family support;an inability to meet basic living needs;a belief their financial crisis will substantially worsen;financial supports being inadequate to meet need;a diminished sense of wellbeing at work and home; andwidespread experiences of racism [2].

During the pandemic, many temporary visa holders lost their jobs and incomes, and many reported experiencing more exploitative working conditions and abuse [2]. More than 70% of temporary migrants reported either job loss or substantially reduced work hours. Many also reported reduced hourly wages (21%), doing unpaid work (11%), being forced to do tasks they were uncomfortable with (13%), and performing work in exchange of food and/or housing rather than wages (13%) [2].

## 8. Exclusion from Welfare, Public Services, and COVID-19 Supports

Prior to the pandemic, people with temporary migrant status in Australia were excluded from all welfare supports and, in most instances, were not entitled to free public services such as healthcare and public education for their children. As this group cannot access welfare and usually must pay for public services, they are at risk of poverty, homelessness, and food insecurity, particularly if they lose their job.

In keeping with the above trend of excluding temporary visa holders from welfare support, Australia explicitly excluded ‘temporary’ migrants from accessing the government’s emergency COVID-19 economic assistance measures. In April 2020, then Prime Minister Scott Morison delivered a ‘go home’ message to people holding temporary visas who experienced job losses and could no longer afford to sustain themselves in Australia. However, it is important to note that Australia’s response towards temporary visa holders was at odds with other countries. The United Kingdom, Canada, New Zealand, Ireland, and other countries recognised temporary migrants as valuable members of society and responded through extending wage subsidies and other forms of support, including unemployment payments and housing support, to temporary migrants residing in these countries [2].

Excluding members of society with temporary migrant status from essential economic aid and support during a global pandemic can be viewed as part of the institutional machinery producing sub-citizenship through drawing sharp divisions between people based on migration and citizenship status [4,16]. Symington (2021) [4] notes that “despite being taxpayers that contribute a crucial sector to the Australian workforce, migrants holding temporary visas were forced in much of Australia to rely on their savings and superannuation and, where those were exhausted or non-existent, charity” (p. 10).

Framing temporary visa holders as undeserving of pandemic response measures compounded pre-existing economic and social inequalities faced by this group. As the government’s Jobkeeper and Jobseeker payments were not made accessible to workers with temporary migrant status, some employers responded through laying off temporary migrant employees en masse whilst retaining people with permanent residency and citizenship status [65,66]. Given that Australia’s COVID-19 response policies penalised businesses that employ temporary migrant workers [10], it is hardly surprising that some employers responded through standing down employees with temporary migrant status in favour of those who were eligible for the government’s COVID-19 economic response payments.

## 9. COVID-19 Response Policy’s Implications for ‘Temporary’ Migrant Men and Families

Having reviewed how Australia’s COVID-19 policies exacerbated pre-existing social and economic insecurities among temporary migrants, we can now consider how widespread job losses and financial hardship alongside exclusions from COVID-19 supports and free public services may impact the health and wellbeing of temporary migrant men and their families.

Prior to the emergence of the COVID-19 pandemic, Wickramage et al. (2018) [59] observed that people residing in nation-states with ‘non-citizen’ status ‘may be disproportionately affected in the event of health emergencies’ due to of a combination of political, sociocultural, economic, and legal barriers that results in their ‘limited access to and awareness of health and welfare services’ (p. 251). As previously discussed, before COVID-19, temporary migrants in Australia were far more vulnerable to experience job, housing, and food insecurity than those who held more secure migration and citizenship statuses. Australia’s pandemic response policies exacerbated the already precarious economic and social situation faced by temporary migrants living in the country, as they were among the first to lose their jobs; were ineligible to access any form of welfare support or free public services, including healthcare and public education; and often have little or no social support networks to rely upon [10,67].

As discussed earlier, there is a dearth of information available on temporary migrant men’s health and wellbeing in general in the context of COVID-19. Through piecing together the available information on migrant men’s work and fathering practices, migrant men’s health and wellbeing, and Berg and Farbenblum’s (2020) [2] recent survey findings of temporary migrants in Australia during COVID-19, we can infer some of the likely post-COVID health and wellbeing implications for men and their families with temporary migration status.

It is widely known that fathers, both migrant and non-migrant, are more likely to be the breadwinners in the family [68,69,70,71], with emerging evidence indicating that many migrant and refugee families heavily rely on the male partner’s income to survive [30]. Migrant and refugee fathers in Australia strongly identify as being providers for their families [30,72,73,74]. Moreover, their sense of wellbeing and capacity for engaged fathering often hinges on fulfilling the breadwinner role [30].

The available data in the Australian context focus specifically on the mental health and wellbeing of migrant and refugee-background fathers with more secure migration and citizenship statuses, (i.e., permanent residents and citizens). The key learnings from a small number of studies that draw upon LSAC data prior to the pandemic suggest that migrant and refugee men’s mental health and wellbeing is strongly influenced by their employment and job conditions [75] and that migrant and refugee fathers, particularly those from non-English speaking countries, are more likely to experience poorer mental and physical health than Australian-born fathers [76].

While, presently, there is no empirical data focused on the health and wellbeing of temporary migrant men in Australia in the context of COVID-19, we can deduce with some degree of confidence from the available evidence that men holding temporary visas are at least as likely to experience similar general health and mental health challenges as those with more secure residency and citizenship statuses. Taking this line of reasoning a step farther, it is likely that ‘temporary’ migrant men are at risk of experiencing more pronounced mental health and general health challenges than those with permanent residency and citizenship status, as migrant men’s employment conditions strongly influence their health and wellbeing.

Alongside the findings above pertaining to the intersections between employment, fathering, and health and wellbeing among migrant and refugee men, one large-scale study focusing on people with temporary migrant status in Australia found that widespread job losses in the context of exclusionary COVID-19 response policies had devastating financial and social impacts upon this group [2]. The findings of this large-scale study demonstrate that temporary migrants in Australia faced increased economic, job, and housing insecurity; unfair and exploitative working conditions; and increased experiences of racism and abuse during the pandemic.

As migrant families often rely upon fathers to survive economically, if a migrant father faces employment challenges or a reduced income, the family falls into a very difficult financial position. The situation is more economically precarious for temporary migrants in Australia as they are ineligible for welfare, COVID-19 supports, and access to free public services, such as healthcare and public education for their children. Being excluded from social safety nets and supports puts additional pressure on ‘temporary’ migrant men to economically support their families.

When migrant men are unable to fulfill the breadwinner role, they often undergo emotional and mental stress and suffer a crisis of masculinity [74,77]. As migrant men on temporary visas are more vulnerable to exploitative work conditions and are ineligible for COVID-19 schemes or other forms of economic and social supports, they have fewer options available to leave unscrupulous employers. Under such conditions, ‘temporary’ migrant men may continue working in jobs where they experience abuse and exploitation as they as they feel compelled to fulfill what they regard as their cultural duty of being financial providers to their families.

## 10. Summary and Conclusions

This paper applied an intersectional sub-citizen lens to explore how Australia’s COVID-19 policy responses multiplied and compounded pre-existing economic and social exclusions faced by temporary migrant men. Migrant men are often the breadwinners who feel a cultural obligation to be financial providers for their families. As temporary visa holders in Australia were more likely to experience employment-related stresses, including job loss, reduced work hours, exploitation, and racism, and were ineligible for COVID-19 welfare and income safety nets, it is likely that temporary migrant men were less able to fulfill the role of being financial providers for their families. Being excluded from welfare safety nets and services, including access to COVID-19 economic supports, is likely to adversely impact the health and wellbeing of ‘temporary’ migrant men and their families.

People holding temporary visas are positioned low in the sub-citizen hierarchy. Due to their classification as ‘non-citizens’, temporary migrants have highly precarious and limited work and residency rights within the nation-state. Through applying a sub-citizenship lens to men with temporary migrant status, this paper explored how construing temporary visa holders as non-citizen outsiders rendered millions of men, women, and children as ineligible for social and economic support in Australia during the global pandemic. 

COVID-19 undoubtedly heightened health, wellbeing, and economic insecurities for people around the world. People with temporary migration status have long been a highly vulnerable group, especially in times of crises. The global pandemic brought to the surface and deepened long-standing structural and legislated inequalities among temporary migrants. These exclusions hinge upon nation-states construing people who hold temporary visas as ‘non-citizen’ outsiders regardless of how long they have lived, worked, and contributed to the country.

Temporary visa holders face numerous intersecting inequalities that are likely to undermine their health and wellbeing at the best of times. During Australia’s COVID-19 recession, temporary migrants experienced disproportionate health, wellbeing, and economic disadvantages, which were compounded as the country opted not to recognise this group as belonging to society and deserving of access to welfare supports, public services, and safety nets.

In the absence of studies focused on temporary migrant men’s health and wellbeing in the context of COVID-19, this paper has pieced together the available global literature on men’s health and COVID-19, migrant men’s health and wellbeing, and the social and economic conditions of temporary migrants in Australia during the pandemic. Viewing these threads of literature though a sub-citizen lens, it contends that exclusionary social and economic policies faced by temporary migrants have been compounded during the pandemic. The combination of Australian immigration and COVID-19 response policies likely undermined the health and wellbeing of temporary migrant men and their families.

Certainly, there is a great need for future empirical research to explore how the health and wellbeing of temporary migrant men and their families have been impacted post-COVID. Comparative studies drawing upon the voices of diverse groups of temporary migrants living in Australia and different countries would be a welcome advancement in this space. Such scholarship would build an evidence base to understand how policy responses impact the health and wellbeing of diverse groups of temporary migrants in different countries and contexts around the world.

Through reinforcing the social boundaries and hierarches between ‘citizens’ and ‘non-citizens’, Australia’s COVID-19 policy responses have deepened pre-existing social and economic exclusions among a highly vulnerable group of people living, often long-term, with precarious residency status in the country. In failing to extend new and existing welfare supports and COVID-19 response measures to temporary migrants and communicating a ‘go home’ message to those who experienced job loss, the country refused to accept responsibility for millions of people who live, pay taxes, contribute to social and economic life, and call Australia their home. We have yet to see what the long-term impact of exclusionary COVID-19 response policies will be on temporary migrant men and families and on the willingness of future immigrants to make Australia their home.

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
