# Peer review of "COVID-19 and Policy-Induced Inequalities: Exploring How Social and Economic Exclusions Impact ‘Temporary’ Migrant Men’s Health and Wellbeing in Australia"

_ijerph, 2023, doi:10.3390/ijerph20136193_

Round 1

Reviewer 1 Report (Previous Reviewer 2)

The manuscript has significantly improved since the last version I reviewed. The authors have improved the contextualization of the problem for global health and the theoretical background for data analysis.

Author Response

Thank you very much for the useful feedback you provided. I believe the revisions that you suggested significantly strengthened the paper. 

Reviewer 2 Report (Previous Reviewer 1)

You mentioned the page 9 as including answers to my question `What is new in this article compared to previous approaches of sub-citizenship theory?` Nothing new was added there.

Author Response

Thank you for your helpful suggestions, which I feel significantly strengthened the paper.

The paper’s application of sub-citizenship theory is new in that it draws upon the theory to understand how political and social processes of subordination affect the health and wellbeing among people with insecure migration status in Australia (i.e. temporary migrant men).

In the previous round of revisions, I addressed the question you posed about the newness of this paper’s application of sub-citizenship theory by adding the following to the paper:

(p.4) To operationalise the critical social policy approach I apply an intersectional sub-citizen lens to explore how Australia’s COVID-19 policy responses multiplied and compounded pre-existing economic and social exclusions faced by people with temporary migration status. Sub-citizenship theory (Rung, 2020) enables a deep consideration of how the predominance of neoliberal dynamics underpinning immigration and public policies works to sort and categorise people to experience different levels of subordination and inequalities largely based upon the allocation of migration and citizenship status. Through a sub-citizenship approach, the focus is on how it has come to be that pre-existing exclusions and vulnerabilities with respect to work and access to welfare supports and public services have become compounded for those with temporary migrant status and how such exclusions are likely to result in adverse health and wellbeing among temporary migrant men.

(p. 9) This paper’s use of sub-citizenship theory focusing on the health and wellbeing implications among temporary migrant men in the context of COVID-19 policy exclusions expands upon previous approaches to sub-citizen that have analysed how policies and practices led to the subordination and dehumanisation of other groups of migrants including children with illegalised migrant status in Australia and America (Rung, 2020).

Then I listed some of the other ways sub-citizenship theory has been applied by other researchers.

Reviewer 3 Report (New Reviewer)

I have pointed out some areas which can be improved. Methods sections needs to be mentioned. 

Thorough spelling check needs to be done. British and American English must not be mixed up. 

Author Response

Thank you for your helpful suggestions. Please see attached files for my responses to your suggestions.

Round 2

Reviewer 3 Report (New Reviewer)

accepted

This manuscript is a resubmission of an earlier submission. The following is a list of the peer review reports and author responses from that submission.

Round 1

Reviewer 1 Report

1. Useful to clarify the status of this article. Is it a research article, a systematic review, or a simple essay using a literature review?

2. In order to explore the barriers  created  by immigration policies of the Australian Government for temporary migrants to integration and to determine the consequences of these barriers for the well-being of immigrants, might be useful to: a) define in an operational way what are the institutional definitions of permanent vs temporary immigrants; b) to measure consequences of these barriers for different categories of immigrants in pandemia crisis; to do multiple comparisons referring to a) and b) dimensions.

3. What is new in this article compared to previous approaches of sub-citizenship theory?

4. Do you have a shred of empirical evidence showing that temporary immigrants are homogenous in terms of the consequences of the government's discriminatory policies by migration status and gender?

5. What are the views of temporary immigrants on discriminatory policies of the Australian government?

6. It would be useful to be in a position to report what are the consequences of the anti-COVID19 measures on vulnerable groups, controlling for socio-demographics.

 Government policies 

Author Response

1. I specified that this is a critical social policy analysis paper and that I use sub-citizenship lens to operationalise this approach (p. 3).
2. A)Added 2 paragraphs (pp. 5-6) explaining the institutional definitions of permanent migrant and temporary migrant.

B)Thank you for your suggestion, but as this paper focuses on the policy implications for men with temporary migrant status, I have opted to focus this paper on the consequences faced by this particular group of people. Outlining the consequences of different categories of migrants during the pandemic and doing multiple comparisons referring to the dimensions would make this paper far too lengthy. However, I have added an explanation of the warrant of the paper’s focus on those with temporary migrant status into the intro and sprinkled a few examples throughout the paper.

3.  This article deploys sub-citizenship theory as a lens to explore how people with temporary migrant status are a particularly vulnerable group due in part from being institutionally excluded from Covid-19 policy responses.

I added more (p.4 & 9) to explain how I used sub-citizenship in this paper and how using it to analyse temporary migrants health and wellbeing experiences in Australia in the post-COVID context is new and compares to others who have used the theoretical lens prior to this paper.

4.   Apologies, it was certainly not my intention to imply that temporary migrants are a homogenous group. Indeed, they are quite diverse. However, the Australian government’s policies towards those with ‘temporary’ migrant status are uniformly exclusionary with respect to accessing services and public safety nets, which in turn presents this group of people with an array of adverse health, wellbeing, social and economic consequences. I added a more in-depth an explanation of this on pp. 2-3.

Throughout the paper I draw upon an array of available empirical evidence including the large-scale survey of temporary migrants (Berg and Farbenblum, 2020), and other academic and gray literature exploring the adverse health and wellbeing issues experienced by temporary migrants in the context of COVID-19 job losses and policy exclusions. In the draw upon 2 threads of literature: COVID-19 and men’s health and temporary migration literature pre and post-COVID. I argue that more scholarship is needed to explore how social and economic policy exclusions affect the health and wellbeing of particular sub-sets of temporary migrants post-COVID.

5.  As this paper is a critical analysis of public policy and not an empirical research article, it does not directly draw upon the voices of people with temporary migrant status. 

I added a sentence (p. 14) calling for future research to do this important work of including the views of temporary migrants to comment and critique the policies that exclude them.

6. Thanks. I agree this would be an interesting and worthwhile exercise for another paper. However, this paper is focused on analysing the likely health and wellbeing consequences for temporary migrant men and their families in the context of exclusion from COVID-19 support policies in Australia. It is beyond the scope of this paper to report on the consequences of COVID-19 exclusionary measures on vulnerable groups controlling for socio-demographics.

Reviewer 2 Report

I reviewed the manuscript “COVID-19 and policy-induced inequalities: Exploring

how social and economic exclusions impact ‘temporary’ migrant men’s health and wellbeing in Australia” for IJERPH.

It is an interesting manuscript, but in my view, it does not add anything to what is already known on the subject.

The approach is generalist and punctual and fails to address important issues to understand the theme, as described below:

1. SDG and Agenda 2030:

COVID-19 has exacerbated inequalities in health care, housing and employment. Even before the pandemic, many minority and vulnerable groups such as lesbian, gay, bisexual, transgender and intersex (LGBTI) refugees faced continued threats after fleeing prison and violence at home. However, the text does not address the central issues related to global health, the SDGs (mainly 3 and 10) and how this impacts on its discussion

In addition, there is a lack of contextualization of policies for temporary immigrants... is there a difference for undocumented immigrants?

2. What makes the male population stand out: For this, I exemplify with the excerpt “The available evidence suggests that being male is a key factor determining the health and wellbeing outcomes of people who contract COVID-19. Sex-disaggregated epidemiological analyzes across different parts of the world reveal a gender gap associated with the virus (Bwire, 2020)”.

The authors bring a paper from 2020 to support this, although the profile has already changed, and the male population is already associated with worse health prognoses.

3. A vulnerability benchmark is missing to explain this blind spot:

It is impossible to understand the results without knowing which theoretical lens the author used. What vulnerabilities are you considering? What's the benchmark? What about overlapping vulnerabilities? How is it addressed? For example, there are several surveys with the male and immigrant population that show that LGBTs had worse health prognoses because they are LGBT, men, immigrants... this must be addressed.

Author Response
1. I’ve added to the intro a little section about central issue related to global health, with reference to SDGs 3 and 10. I’ve also tied this back into the conclusion.

The United Nations Sustainable Development Goals (SDG) can be advanced through appropriate and inclusive public health, immigration, and social policies that promote good health and wellbeing (SDG 3) and are aimed at reducing inequality (SDG 10) (United Nations, 2023). With these goals in mind, this paper aligns with scholarship advocating for a gendered and human rights lens in public health and pandemic research and policy responses. (p. 4)

Excluding certain groups of people from COVID-19 policy responses takes us farther from realising the United Nations goals of ensuring health lives and promoting wellbeing for all people and reducing inequality within and between countries (p.14).

The health and wellbeing inequalities experienced by undocumented migrants are considerably worse in Australia as this group is institutionally positioned at a far more precarious level of sub-citizenship, often without work rights and at risk of being held in detention (see Rung, 2020). I’ve added more to explain that while this paper does not delve into undocumented migrants, I acknowledge that different groups of migrants are positioned at different levels of the sub-citizen hierarchy, which presents them with different levels of social, economic, and health exclusions (p. 3).

It must also be acknowledged that certain groups of migrants, including undocumented migrants, are institutionally positioned at lower levels of the sub-citizenship hierarchy making them more at risk of experiencing social, economic and health inequalities. For example, Australia’s punitive immigration detention regime confines undocumented migrants in close quarters in detention centres where they faced extremely high risks of COVID-19 infection and possibly death (Vogl, Flay, Loughnan, Murray, and Dehn, 2021).

2. I explained the warrant for focusing on the male population of temporary migrants by drawing upon 2 threads of literature: men’s health and COVID-19 and academic and gray literature focused on the temporary migrant’s health and wellbeing pre- and post-COVID. I added more to the paper to strengthen this warrant and I point out that there is a need for more empirical studies to focus on diverse sub-sets of migrants, including temporary migrant men in a post-COVID context.

3. Sub-citizenship theory is the theoretical lens used. I have elaborated more on how I’ve applied the theory:

This paper applied an intersectional sub-citizen lens to explore how Australia’s COVID-19 policy responses multiplied and compounded pre-existing economic and social exclusions faced by people with temporary migration status, with particular focus on men. (p.14)

I’ve drawn upon intersectionality to assess overlapping and multiple exclusions that can make men with temporary migrant status a particularly vulnerable group in the country in the context of COVID-19 policy exclusions.

I addressed the issue you raised about LGBT migrants in the following statement (p.3):

This paper’s focus on men holding temporary visas should not be taken as a statement they constitute a homogenous group nor that they face more structural inequalities than women and LGBT communities who hold temporary visas. People of all genders have been, and will likely continue to be, affected by COVID-19 in diverse ways for years to come. While this paper primarily focuses on men with temporary migrant status, many of the issues raised also apply to women, children and TGNB communities, albeit in complex and varied ways.

I added the following acknowledging the existence of overlapping vulnerabilities:

Indeed, emerging research suggests that LGBT migrant populations face overlapping social and political factors that structure health inequalities (Kline, 2020). While this paper’s intersectional approach focuses on men with temporary migrant status, it acknowledges that certain sub-groups of temporary migrant men, including LGBT migrants, experience compounded forms of social, economic, and health inequalities.